# Relationships Among and Predictive Values of Obesity, Inflammation Markers, and Disease Severity in Pediatric Patients with Obstructive Sleep Apnea Before and After Adenotonsillectomy

**DOI:** 10.3390/jcm9020579

**Published:** 2020-02-20

**Authors:** Hai-Hua Chuang, Chung-Guei Huang, Li-Pang Chuang, Yu-Shu Huang, Ning-Hung Chen, Hsueh-Yu Li, Tuan-Jen Fang, Jen-Fu Hsu, Hsin-Chih Lai, Jau-Yuan Chen, Li-Ang Lee

**Affiliations:** 1Department of Family Medicine, Linkou Chang Gung Memorial Hospital, Taoyuan 33305, Taiwan; chhaihua@gmail.com (H.-H.C.); welins@cgmh.org.tw (J.-Y.C.); 2Department of Family Medicine, Chang Gung University, Taoyuan 33302, Taiwan; 3Department of Industrial Engineering and Management, National Taipei University of Technology, Taipei 10608, Taiwan; 4Obesity Institute & Genomic Medicine Institute, Geisinger, Danville, PA 17822, USA; 5Department of Laboratory Medicine, Linkou Chang Gung Memorial Hospital, Taoyuan 33305, Taiwan; joyce@cgmh.org.tw (C.-G.H.); hclai@mail.cgu.edu.tw (H.-C.L.); 6Department of Medical Biotechnology and Laboratory Science, Graduate Institute of Biomedical Sciences, Chang Gung University, Taoyuan 33302, Taiwan; 7Department of Pulmonary and Critical Care Medicine, Sleep Center, Linkou Chang Gung Memorial Hospital, Taoyuan 33305, Taiwan; lpchuang1678@yahoo.com.tw (L.-P.C.); ninghung@cgmh.org.tw (N.-H.C.); 8Faculty of Medicine, Graduate Institute of Clinical Medicine Sciences, Chang Gung University, Taoyuan 33302, Taiwan; yushuhuang1212@gmail.com (Y.-S.H.); hyli38@cgmh.org.tw (H.-Y.L.); fang3109@cgmh.org.tw (T.-J.F.); jeff0724@gmail.com (J.-F.H.); 9Department of Child Psychiatry, Sleep Center, Linkou Chang Gung Memorial Hospital, Taoyuan 33305, Taiwan; 10Department of Otorhinolaryngology-Head and Neck Surgery, Sleep Center, Linkou Chang Gung Memorial Hospital, Taoyuan 33305, Taiwan; 11Department of Pediatrics, Linkou Chang Gung Memorial Hospital, Taoyuan 33305, Taiwan; 12Research Center of Bacterial Pathogenesis, Chang Gung University, Taoyuan 33302, Taiwan; 13Institute of Brain Science, National Yang-Ming University, Taipei 11221, Taiwan

**Keywords:** adenotonsillectomy, basic fibroblast growth factor, children, granulocyte-macrophage colony-stimulating factor, interleukin, monocyte chemotactic protein, obesity, obstructive sleep apnea, platelet-derived growth factor, RANTES

## Abstract

Both obstructive sleep apnea (OSA) and obesity are major health issues that contribute to increased systemic inflammation in children. To date, adenotonsillectomy (AT) is still the first-line treatment for childhood OSA. However, the relationships among and predictive values of obesity, inflammation, and OSA severity have not been comprehensively investigated. This prospective study investigated body mass index (BMI), serum inflammatory markers, and OSA severity before and after AT in 60 pediatric patients with OSA. At baseline, differences in levels of interleukin-6, interleukin-9, basic fibroblast growth factor, platelet-derived growth factor-BB, as well as regulated on activation, normal T cell expressed and secreted (RANTES) were significant among the various weight status and OSA severity subgroups. After 3 months postoperatively, the differences in these inflammatory markers diminished along with a decrease in OSA severity while obesity persisted. The rate of surgical cure (defined as postoperative obstructive apnea-hypopnea index < 2.0 and obstructive apnea index < 1.0) was 62%. Multivariate analysis revealed that age, BMI z-score, granulocyte-macrophage colony-stimulating factor, monocyte chemotactic protein-1, and RANTES independently predicted surgical cure. Despite the significant reductions in inflammatory markers and OSA severity after AT, an inter-dependent relationship between obesity and OSA persisted. In addition to age and BMI, several inflammatory markers helped to precisely predict surgical cure.

## 1. Introduction

Obstructive sleep apnea (OSA), a common chronic disorder with an increasing prevalence in many developed countries [1,2,3,4], is characterized by recurrent episodes of partial and complete airway obstructions during sleep with repetitive apneas and hypopneas [2]. OSA is associated with various co-morbidities that affect multiple organ systems, resulting in acute events or long-term sequelae, and consequently incurring considerable social and economic costs [2]. Previous studies have shown that obesity is one of the most important risk factors for OSA, along with increasing age, male sex, and craniofacial structure abnormalities [2,3].

Among those affected, pediatric patients are of particular concerns. Pediatric OSA is associated with neurobehavioral, cognitive, cardiovascular, as well as metabolic morbidities. The prevalence of the disease is reported to be 2%–3%. Enlarged lymphoid tissues such as the tonsils and adenoids are considered the most important factors for pediatric OSA [5]. Obesity seems to play a role too, however, no sufficient evidence has been reported to support a strong effect in children as being reported in adults [6,7,8]. Considering the lack of solid links between weight status, lymphoid tissue size, and OSA severity in pediatric patients, other pathophysiological mechanisms have been postulated [9,10], either alone or in combination with obesity. 

Systemic inflammation is one of the most commonly proposed mechanisms, and it has been suggested to influence OSA severity and associated co-morbidities [11,12]. A study in Israel reported enhanced somatic growth after adenotonsillectomy (AT) in association with a decrease in systemic inflammation and an increase in caloric intake, showing the co-existence of systemic inflammation and OSA-related morbidities [13].

AT has been reported to significantly improve indices of sleep-disordered breathing and remains the recommended first-line treatment [14,15]. However, not every pediatric patient benefits from this procedure [14,16], and the OSA resolution rate after surgery is sub-optimal. Moreover, some patients have better outcomes than others, and the factors that promote incomplete resolution of OSA after surgical treatment have yet to be clearly defined [15,16]. Therefore, a pre-surgical evaluation tool that can better predict post-surgical outcomes is urgently needed and will have great clinical application value in the field.

The first aim of this study was to understand the relationships between age, sex, weight status, disease severity, and inflammation level in pediatric OSA patients. The second aim was to develop a predictive model for the surgical cure rate using both conventional well-known factors and inflammatory biomarkers.

## 2. Method

### 2.1. Ethical Considerations

This study was approved by the Institutional Review Board of the Chang Gung Memorial Foundation (104-7279A3). Written informed consent was obtained from all parents and participants ≥6 years.

### 2.2. Participants

We prospectively recruited consecutive children who were referred to the Department of Otorhinolaryngology-Head and Neck Surgery at Linkou Chang Gung Memorial Hospital (Taoyuan, Taiwan) for AT between March 1, 2017 and January 31, 2019. The primary inclusion criteria were: (a) age 5–12 years; (b) obstructive apnea-hypopnea index (OAHI) ≥ 5.0 events/h or OAHI ≥ 1.0 event/h plus at least one OSA-related morbidity (such as elevated blood pressure, daytime sleepiness or learning problems, growth failure and enuresis) [17,18,19]; and (c) agreeing to answer the OSA-18 questionnaire [20] and to undergo blood sampling and/or polysomnography at baseline and after AT. The exclusion criteria were: (a) refusing to participate after instruction; (b) being unable to undergo general anesthesia and AT after preoperative evaluations; and (c) craniofacial, neuromuscular, or chronic inflammatory disorders such as asthma, allergies, eczema, or other atopic/autoimmune disease [21].

### 2.3. Polysomnography

We used standard in-laboratory full-night polysomnography with simultaneous video recording to document OSA parameters at baseline and 3 months after AT. During all nocturnal polysomnography recordings, a family member was required to be present [19]. The AHI was defined as the sum of all obstructive and mixed apneas (≥ 90% decrease in airflow for a duration of ≥ 2 breaths), plus hypopneas (≥ 50% decrease in airflow and either ≥ 3% desaturation or electroencephalographic arousal, for a duration of ≥ 2 breaths), divided by the number of hours of total sleep time [22]. In this study, we further focused on a subgroup of patients with OAHI ≥ 2.0 events/h or obstructive apnea index (OAI) ≥ 1.0 events/h according to a key reference study [23]. Thereafter, the OAHI, OAI, mean oxygen saturation measured by pulse oximetry (SpO_2_), and minimal SpO_2_ were recorded for further comparisons. In the present study, the patients were categorized as having ‘severe’ (OAHI ≥ 10.0 events/h) or ‘non-severe’ (OAHI ≥ 2.0 events/h to < 10.0 events/h) OSA [24]. ‘Surgical cure’ was defined by a reduction in both the OAHI < 2.0 events/h and the OAI < 1.0 event/h (resolution of OSA) after AT [23]. The subjects were further divided into four subgroups based on their weight status and OSA severity: ‘non-obese with non-severe OSA’ (‘nO-nS’), ‘non-obese with severe OSA’ (‘nO-S’), ‘obese with non-severe OSA’ (‘O-nS’), and ‘obese with severe OSA’ (‘O-S’). In this study, ‘obesity’ was defined as a body mass index (BMI) z-score ≥1.645 [25].

### 2.4. Measurement of Inflammatory Biomarkers

Serum was separated from the whole blood morning samples drawn (clotting time = 30 min) from each participant and stored at −80℃ until assay. Patients with acute systemic inflammation were not tested until the conditions had abated [26]. The concentrations of 27 inflammatory biomarkers were determined using the Bio-Plex^®^ Pro Human Cytokine 27-plex panel (Bio-Rad Laboratories, Hercules, CA, USA). Specifically, the following analytes were measured: interleukin (IL)-1β, IL-1 receptor antagonist (IL-1ra), IL-2, IL-4, IL-5, IL-6, IL-7, IL-8, IL-9, IL-10, IL-12, IL-13, IL-15, IL-17, basic fibroblast growth factor (basic-FGF), eotaxin, granulocyte-colony stimulating factor (G-CSF), granulocyte-macrophage colony-stimulating factor (GM-CSF), interferon-γ, interferon gamma-induced protein (IP)-10, monocyte chemotactic protein (MCP)-1, macrophage inflammatory protein (MIP)-1α, MIP-1β, platelet-derived growth factor (PDGF)-BB, regulated on activation, normal T cell expressed and secreted (RANTES), tumor necrosis factor (TNF)-α, and vascular endothelial growth factor (VEGF). All samples were centrifuged for 20 s at 14300 revolutions per minute to remove debris. Fifty microliters of each sample was diluted at a ratio of 1:3 in sample diluent and then incubated with antibody-coupled beads for 60 min to allow binding. They were further incubated with detection antibodies for 30 min, and then conjugates were treated with streptavidin for 10 min. The reagents in each step were washed on a Bio-Plex Pro II wash station. Duplicate measurements of all samples were analyzed using a Bio-Rad Bio-Plex Luminex 200 instrument (Hercules, CA, USA) and the Bio-Rad Bio-Plex Manager software (v6.0; Hecules, CA, USA).

### 2.5. AT

Using the plasma knife technique (PEAK PlasmaBlade; Medtronic Inc, Jacksonville, FL, USA), the principal investigator (L.A.L.) performed all surgeries under general anesthesia with an average hospitalization of 2 days. The detailed surgical techniques have been described elsewhere [27].

### 2.6. Statistical Analysis

Means and standard deviations were used to summarize continuous variables, and numbers with percentages were used to present categorical variables. Student’s *t*-tests and one-way analysis of variance (ANOVA) with post-hoc Tukey’s honestly significant difference tests were used to compare continuous variables, and Fisher’s exact tests and chi-square tests were used to compare categorical variables in different groups, as appropriate. Pearson correlation test was used to analyze the association between continuous variables whereas Spearman correlation test was used to analyze the association between categorized variables and continuous variables. Paired Student’s *t*-tests were used to compare changes in the variables 3 months postoperatively. Variables with a *p*-value < 0.05 in the Student’s *t* and ANOVA tests were further dichotomized according to the optimal cut-off value using receiver operating characteristic curves [28]. The dichotomized variables were then assessed using multivariate logistic regression models. All *p*-values were two-sided, and statistical significance was accepted at *p* < 0.05. All statistical analyses were performed using SPSS software (version 23; International Business Machines Corp., Armonk, NY, USA).

## 3. Results

### 3.1. Patients’ Characteristics at Baseline

A total of 60 Taiwanese children from Han ancestry with OSA (14 [23%] girls and 46 [77%] boys) with a mean age of 7.5 ± 2.2 years completed follow-up assessments. The patients’ characteristics in the four subgroups categorized by weight status and OSA severity at baseline are shown in Table 1. As expected, there were significant differences in BMI z-scores, OAHI, OAI, mean SpO_2,_ and minimal SpO_2_; however, there were no significant differences in age, sex, and OSA-18. 

The nO-nS subgroup had lower OAHI (compared to nO-S, O-S), lower OAI (compared to nO-S), higher minimal SpO_2_ (compared to O-S), and higher minimal SpO_2_ (compared to nO-S, O-S). The nO-S subgroup had higher OAHI (compared to nO-nS, nO-S), higher OAI (compared to nO-nS, nO-S), and lower minimal SpO_2_ (compared to nO-nS, O-nS). The O-nS subgroup had lower OAHI (compared to nO-S, O-S), lower OAI (compared to nO-S), higher minimal SpO_2_ (compared to O-S), and higher minimal SpO_2_ (compared to nO-S, O-S). The O-S subgroup had higher OAHI (compared to nO-nS, O-nS), lower minimal SpO_2_ (compared to nO-nS, O-nS), and lower minimal SpO_2_ (compared to nO-nS, O-nS).

### 3.2. Inflammatory Biomarkers at Baseline

Table 2 shows the levels of the inflammatory biomarkers across obesity and OSA severity subgroups at baseline. The average time duration of the sleep study and collection of the markers of inflammation at baseline was 3.9 ± 0.7 weeks. Differences in the levels of IL-6 (Figure 1d), IL-9 (Figure 1e), basic-FGF (Figure 1f), PDGF-BB (Figure 1g), and RANTES (Figure 1h) among the four subgroups were significant. The nO-nS subgroup had a significantly lower level of IL-6 than the O-S subgroup and significantly higher levels of IL-9 and basic-FGF than the O-nS subgroup. Furthermore, the nO-S subgroup had significantly higher levels of PDGF-BB and RANTES than the O-nS subgroup.

### 3.3. Associations between Patients’ Characteristics and Inflammatory Biomarkers at Baseline

The associations of age with the levels of basic-FGF (*r* = −0.32, *p* = 0.01) and interferon-γ (*r* = −0.29, *p* = 0.02) were significant. The BMI z-score was significantly related to the level of IL-8 (*r* = −0.31, *p* = 0.01). The sex, OAHI, OAI, mean SpO_2_, and minimal SpO_2_ were not statistically significantly correlated with the levels of inflammatory biomarkers (data not shown).

### 3.4. Patients’ Characteristics After AT

The mean follow-up period after AT was 4.8 ± 2.0 months. Overall, the mean BMI z-score increased from 0.58 ± 2.05 to 0.85 ± 1.5 (*p* = 0.08), the mean OSA-18 score reduced from 80.7 ± 15.6 to 51.8 ± 13.8 (*p* = 0.001), and the mean OAHI and OAI significantly reduced from 11.5 ± 13.6 to 2.4 ± 3.1 (*p* <0.001) and from 2.6 ± 6.0 to 0.3 ± 0.7 (*p* = 0.01), respectively. The rate of surgical cure was 62% (37/60). The mean SpO_2_ and minimal SpO_2_ significantly increased from 97.2 ± 1.4% to 97.7 ± 0.8 (*p* = 0.01) and from 87.9 ± 70.7 to 90.9 ± 3.6 (*p* = 0.001), respectively. 

Table 3 shows the postoperative characteristics of the patients across subgroups. The difference in the BMI z-score and OAHI across the four subgroups remained significant (Figure 1a). Postoperative OAHI was significantly lower in the nO-nS subgroup than in the O-nS subgroup (Figure 1b). The rates of surgical cure, in descending order, were 82% for the nO-S subgroup, 75% for the nO-nS subgroup, 46% for the O-S subgroup, and 36% for the O-nS subgroup; the difference in surgical cure rate was significant. Postoperative OSA-18 score, OAI (Figure 1c), mean SpO_2_, and minimal SpO_2_ were equivalent.

Furthermore, the OSA-18 score, OAHI, OAI, mean SpO_2_, and minimal SpO_2_ significantly improved after AT in the nO-nS subgroup. In the nO-S subgroup, the BMI z-score and minimal SpO_2_ significantly increased whereas the OSA-18 score and OAHI significantly reduced after AT. Of note, only the OSA-18 score significantly reduced after AT in the O-nS subgroup. Otherwise, the OSA-18 score, OAHI, OAI, mean SpO_2_, and minimal SpO_2_ significantly improved after AT in the O-S subgroup.

### 3.5. Inflammatory Biomarkers After AT

Overall, postoperative levels of IL-1β (0.3 ± 0.2 vs 0.8 ± 0.8, *p* < 0.001), IL-1ra (113.3 ± 73.5 vs 168.1 ± 137.7, *p* = 0.003), IL-2 (1.9 ± 1.4 vs 3.1 ± 2.2, *p* < 0.001), IL-4 (1.9 ± 1.0 vs 2.7 ± 1.3, *p* < 0.001), IL-5 (6.0 ± 11.6 vs 12.3 ± 15.8, *p* = 0.004), IL-8 (4.1 ± 2.2 vs 6.4 ± 3.6, *p* < 0.001), IL-10 (0.4 ± 0.2 vs 1.2 ± 2.1, *p* = 0.01), IL-15 (24.1 ± 43.6 vs 44.6 ± 48.1, *p* = 0.004), IL-17 (11.3 ± 6.9 vs 15.9 ± 9.4, *p* < 0.001), eotaxin (47.3 ± 21.5 vs 66.6 ± 38.6, *p* < 0.001), GM-CSF (0.8 ± 1.8 vs 1.4 ± 1.8, *p* = 0.02), IP-10 (610.8 ± 398.3 vs 1161.9 ± 2034.1, *p* = 0.04), MCP-1 (18.0 ± 9.5 vs 37.0 ± 26.2, *p* < 0.001), MIP-1α (1.5 ± 1.3 vs 1.9 ± 1.1, *p* = 0.03), MIP-1β (118.3 ± 22.8 vs 128.2 ± 30.6, *p* = 0.01), PDGF-BB (5072.9 ± 62246.0 vs 6816.0 ± 2888.7, *p* < 0.001), and RANTES (16617.9 ± 6033.1 vs 22158.3 ± 10292.3, *p* < 0.001) were significantly lower than those measured at baseline. The average time duration of the sleep study and collection of the markers of inflammation during follow-up was 3.7 ± 0.7 weeks, respectively. The average time duration at baseline was equal to that during follow-up (*p* = 0.15). 

There were no significant differences in the postoperative levels of inflammatory biomarkers across subgroups after AT (Table 4). In the nO-nS subgroup, levels of IL-1β, IL-1ra, IL-2, IL-4, IL-5, IL-8, IL-10, IL-13, IL-15, IL-17, eotaxin, GM-CSF, IP-10, MCP-1, MIP-1α, MIP-1β, PDGF-BB (Figure 1g), and RANTES (Figure 1h) significantly decreased compared to baseline. In the nO-S subgroup, levels of IL-1ra, IL-4, IL-17, eotaxin, G-CSF, IP-10, MCP-1, PDGF-BB, and RANTES significantly decreased compared to baseline. In the O-nS subgroup, the level of IL-15 significantly decreased compared to baseline. In the O-S subgroup, levels of IL-1β, IL-1ra, IL-4, eotaxin, G-CSF, MCP-1, and RANTES significantly decreased compared to baseline.

### 3.6. Predictors and Prediction Models for Surgical Cure

We examined receiver operator curves for the prediction of AT surgical cure using single physiological or inflammatory biomarkers (Table 5). The best performers included age < 7.0 years old, BMI z-score < 1.44, IL-1ra < 149.2 pg/mL, IL-9 > 82.8 pg/mL, IL-10 < 0.5 pg/mL, IL-15 < 11.8 pg/mL, IL-17 > 4.7 pg/mL, GM-CSF < 0.2 pg/mL, MCP-1 < 51.2 pg/mL, RANTES > 15435.5 pg/mL, and VEGF < 42.1 pg/mL.

Using multivariate logistic regression analysis, we constructed several predictive models including clinical, inflammatory, and mixed models (Table 6). 

The clinical model included age < 7.0 years old and BMI z-score < 1.44 as independent predictors of AT surgical cure. Accordingly, the physiological model achieved an AUC of 0.79 (*p* < 0.001) with a sensitivity of 95% and specificity of 52% (≥ 1 predictor) (Figure 2a). 

The inflammatory model included IL-1ra < 149.2 pg/mL, IL-17 > 4.7 pg/mL, GM-CSF < 0.2 pg/mL, and MCP-1 < 51.2 pg/mL as independent predictors to predict surgical cure. Accordingly, the inflammatory model achieved an AUC of 0.89 (*p* < 0.001) with a sensitivity of 87% and specificity of 87% (≥ 3 predictors) (Figure 2b). 

Mixed model-1 initially included all statistically significant predictors; the final model included age < 7.0 years, GM-CSF < 0.2 pg/mL, MCP-1 < 51.2 pg/mL, and RANTES > 15435.5 pg/mL to independently predict surgical cure. Accordingly, the mixed model-2 achieved an AUC of 0.92 (*p* < 0.001) with a sensitivity of 87% and specificity of 96% (≥ 2 predictors) (Figure 2c). 

Mixed model-2 combined the clinical model and two of the statistically significant inflammatory biomarkers; the best predictive model included age < 7.0 years, BMI z-score < 1.44, MCP-1 < 51.2 pg/mL, and RANTES > 15435.5 pg/mL to independently predict surgical cure. Accordingly, the mixed model-2 achieved an AUC of 0.91 (*p* < 0.001) with a sensitivity of 92% and specificity of 78% (≥3 predictors) (Figure 2d).

## 4. Discussion

At baseline, differences in BMI z-scores, OAHI, OAI, mean SpO_2,_ and minimal SpO_2_ across subgroups were a result of patient allocation and thus within expectation. There were more boys than girls in every group of children with OSA in this study, which was consistent with previous reports [29]. With regards to inflammatory biomarkers, there were significant differences in IL-6, IL-9, basic-FGF, PDGF-BB, and RANTES across subgroups.

IL-6 is a myokine produced and released from muscle fibers in response to exercise, and it has been shown to have extensive anti-inflammatory functions [30,31,32,33,34,35]. However, IL-6 has also been shown to stimulate inflammatory processes as a pro-inflammatory cytokine in a wide range of diseases including cancers and metabolic, cardiovascular, neurologic, and autoimmune diseases. In the present study, the O-S subgroup had the significantly highest level of IL-6. This could be explained by previous reports of a positive association between IL-6 level and the presence of OSA [36] and also between IL-6 level and obesity and insulin resistance [37]. There was no significant difference between the nO-S and O-nS subgroups, possibly as a result of risk factors in the two groups counteracting each other. In addition, the lack of differences between the nO-nS and nO-S subgroups may have been due to the small sample size of the nO-S subgroup (*n* = 11).

IL-9 is a cytokine secreted by CD4+ helper cells and stimulates cell proliferation and prevents apoptosis [38]. It is encoded by the human IL-9 gene, a candidate gene for asthma [39]. Also, some studies discover IL-9 as a determining factor in the pathogenesis of airway hyper-responsiveness [38]. The literature reports bronchial asthma as an important bidirectional contributing factor for pediatric OSA [40,41,42]. If we regarded IL-9 as somehow a surrogate indicator of airway hyper-responsiveness, then the baseline differences across subgroups were interesting. When comparing subgroups with the same OSA severity, non-obese patients had higher levels of IL-9. We speculated that this was reflection of a greater contribution from airway hyper-responsiveness to OSA in non-obese patients, while obesity contributed less to OSA in this subgroup. The difference was only trendy but not significant between some subgroups though, probably again due to the lack of a bigger sample size.

Basic-FGF, also known as FGF2, is a growth factor and signaling protein in the family of FGFs. FGFs are involved in biological processes including cellular proliferation, survival, migration, and differentiation. Of the family members, basic-FGF is believed to have effects on angiogenesis and adipogenesis [43]. Interestingly, a previous study in Japan reported a negative association between serum basic-FGF levels and BMI [44], while another study in China reported the opposite findings [45]. The observations in our study were consistent with the study from Japan, however, this negative association was observed only in non-severe subgroups and not the severe subgroups. A possible explanation is that in patients with severe OSA, disease severity confounded the effects of obesity and disease. Another explanation may be that an association was not demonstrated due to the small sample size.

PDGF is one of the earliest identified growth hormones. It is primarily synthesized, stored, and released by platelets, but also has other origins including smooth muscle cells, activated macrophages, and endothelial cells. PDGF-BB is one of the five isoforms of PDGF, comprised of two B subunits (PDGFB). PDGF-BB represents one of the strongest mitogens and chemokines in pulmonary arterial smooth muscle cells and adventitial fibroblasts. It plays a crucial role in vascular development and remodeling [46]. A study reported that the serum level of PDGF-BB was higher in OSA patients, with a positive association with AHI and percentage of time spent when oxygen saturation is lower than 90%, as well as a negative association with an average saturation of blood oxygen and lowest saturation of blood oxygen [47]. In our study, other than an expected upward trend (not significant) with higher disease severity, PDGF-BB level was highest in the nO-S and lowest in the O-nS. Patients who were not obese but severe in OSA, tended to have more profound consequences regarding local airway responses.

RANTES, also known as chemokine ligand 5 (CCL5), is classified as a chemokine and is chemotactic for T cells, eosinophils, and basophils, recruiting leukocytes into inflammatory sites [48]. Other than its widely known role in human immunodeficiency virus infection, RANTES has been shown to play a role in inflammation in the liver [49] and upper airway [50]. RANTES has also been reported to be crucially involved in intermittent hypoxia (IH)-induced pre-atherosclerotic remodeling. A previous study observed that systemic inflammation induced by IH, a main component of OSA, was associated with early and predominant RANTES/CCL5 alterations in mice, thereby contributing to IH-induced pre-atherosclerotic remodeling [51]. Another study reported an independent positive association between RANTES level and AHI in adults [52]. 

The patterns of differences across subgroups on PDGF-BB and RANTES displayed interesting and amazing similarities. While obesity and inflammation are the two most important factors in OSA patients, their degree of contribution may differ in adults and children. It is well known that obesity does not play as important a role in the development of OSA co-morbidities in children as it does in adults and that there are other attributable mechanisms. Based on our observations from the study, we speculate that for pediatric patients exhibiting similar OSA severity but with less contribution from obesity, it is possible that their local and systemic inflammatory burdens are even greater than those in their obese peers. This is a very important finding because it suggests how subgroups of pediatric OSA patients may possess different pathogenesis and physiological responses of the diseases, and therefore in need of treatments with different priorities and strategies.

At follow-up at least 3 months after AT, weight status for the overall group and its difference across subgroups remained similar to those at baseline. Disease severity-related parameters improved significantly for the overall group, and differences across subgroups largely diminished. The result suggested that AT seemed to have a profound effect on disease severity, but not weight status. Also, the surgical cure rate tended to be better for the non-obese and more severe patients. As for biomarkers, 19 of the 27 inflammatory markers were significantly reduced after AT, implying an alleviation in inflammatory burdens. The difference across subgroups no longer existed, but the improvement tended to be less significant in the O-nS subgroup. Moreover, residual OSA severities of the O-nS and O-S subgroup were similar (Figure 1a), indicating that there was an inter-dependent relationship between obesity and OSA in obese pediatric patients with OSA.

Based on the literature, the short-term use of continuous positive airway pressure therapy did not seem to improve inflammatory or oxidative biomarkers in OSA patients. Neither did physical activities [53,54]. Our study demonstrated that AT as a procedure to correct adenotonsillar hypertrophy, improved disease severity as well as local and systemic inflammation, without an obvious improvement in obesity. These findings indicated some degree of inflammation caused by adenotonsillar hypertrophy and concomitant OSA, which was supported by a recent study [55]. However, despite being an effective and first-line treatment, the disease resolution rate of AT remained suboptimal [15,16]. The main reason for this is, OSA is a complicated chronic condition with multiple pathogenic factors and co-morbidities. Patients with different phenotypes may differ not only in clinical presentations, but also in pathogenic factors, genetic backgrounds, and disease consequences. This means that the most proper treatment for each patient subgroup may differ.

Based on literature reviews and several of our findings discussed above, we postulate that, given similar disease severity, non-obese pediatric OSA patients, compared to their obese peers, may be stronger in other underlying pathogenic factors, which co-existed with stronger systemic inflammation, as a mediator or a result. These other pathogenic factors could be, for instance, local anatomical structures, central respiratory drives, the ability to utilize oxygen, or susceptibility of hypoxia. As AT is more able to correct these other pathogenic factors than to reduce weight, non-obese pediatric OSA patients are more likely to benefit from their obese peers, in terms of achieving an alleviation in both disease severity and systemic inflammation. This hypothesis is preliminary, yet important and considerable, since it could offer a direction to a better understanding of which and how pediatric OSA patients would benefit from AT. Further investigations would be of great interest.

Last but not least, our study aimed to provide a tool that allowed doctors to be more selective and predictive of their surgical treatments for pediatric OSA patients prior to the procedure. We discovered a few parameters significantly related to surgical cure. Clinical or inflammatory parameters alone did not work as satisfactorily as the mixed models did. The best predictive models included age < 7.0 years, BMI z-score < 1.44, GM-CSF < 0.2 pg/mL, MCP-1 < 51.2 pg/mL, and RANTES > 15,435.5 pg/mL. Previous studies have reported that age [16,56], BMI z-score [16,56], and AHI [16,28,56,57] are predictors of post-AT AHI, in addition to asthma [16], OSA-18 [28], and snoring sound energy [28]. However, GM-CSF, MCP-1, and RANTES as a surgical outcome predictor have never been reported. RANTES, as discussed formerly in the article, is somehow linked to underlying pathogenetic factors, and thus not surprisingly linked to the response of surgical treatment. On the other hand, GM-CSF and MCP-1, which are not significantly associated with weight status or disease severity (data not shown) in our study, are two surprising discoveries concerning outcome prediction. To our best knowledge, there has not been a previous report on relating these two parameters to treatment response. Further studies are warranted to further investigate its clinical application on surgical success prediction.

The main contributions of this study include: (a) reviewing differences in a wide range of inflammatory markers across patient subgroups with different weight status and disease severity, (b) understanding improvements in symptoms and inflammatory burdens across patient subgroups after AT, and (c) developing a predictive model for surgical cure with reasonable performance. The limitations of this study are: (a) selection bias might exist concerning a predominance of boys in our study population, a single Han race of subjects, and a lack of normal control group, which might limit the generalizability on the study; and (b) only short-term outcomes, but not long-term outcomes were assessed in this study. Future research should investigate the long-term effects of obesity and inflammatory biomarkers on OSA severity in studies with a larger sample size to minimize patient bias.

## 5. Conclusions

In summary, pediatric OSA patients with different disease severity and weight status demonstrated different degrees of inflammation. Levels of IL-6 were higher in the obese and severe OSA patients. Levels of IL-9 were higher in non-obese patients than obese patients with the same OSA severity. Basic-FGF was negatively associated with BMI in non-severe patients. PDGF-BB and RANTES had very similar patterns: higher in the non-obese patients and tended to increase with disease severity. The overall group benefited from AT in terms of disease severity-related indices and most inflammatory biomarkers. The surgical cure rate tended to be better in the non-obese and more severe patients. We hypothesize that the young pediatric patients with a higher disease severity benefited most from the procedure because AT corrected the underlying causes of disease more than in the other subgroups. However, a combination of age, BMI z-score, MCP-1, and RANTES, or a combination of age, GM-CSF, MCP-1, RANTES could more accurately predict the resolution of OSA after AT. Further studies on different pathophysiological mechanisms for different phenotypes are of interest in the future and have great clinical application value for developing precise treatments for pediatric OSA patients. 

## Figures and Tables

**Figure 1 jcm-09-00579-f001:**
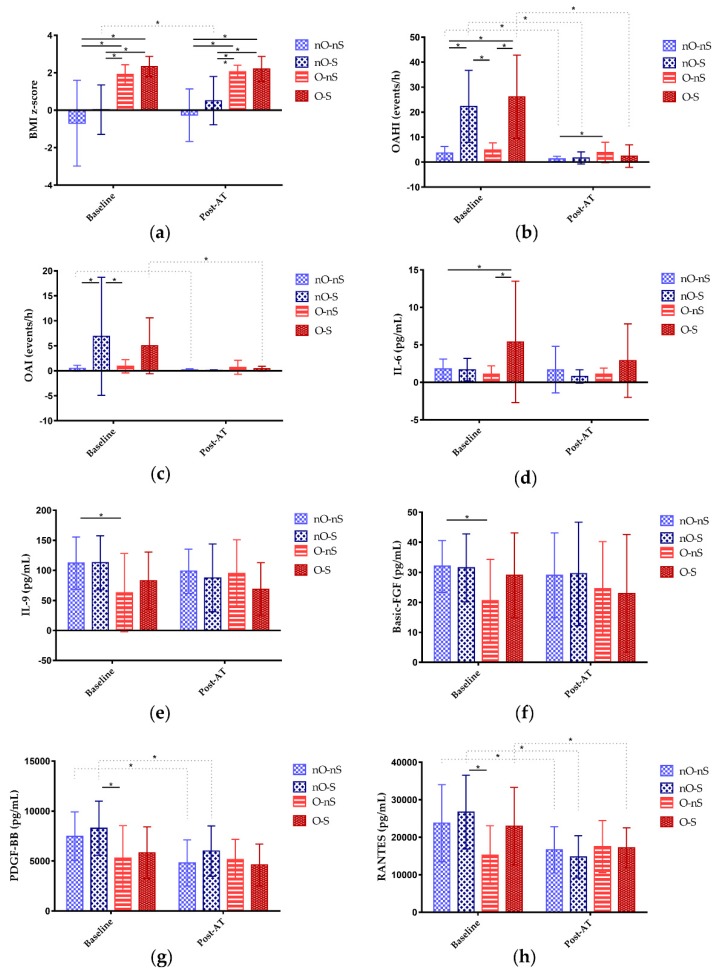
Distributions of variables of interests (means ± standards deviations) at baseline and after adenotonsillectomy (AT). (**a**) Body mass index (BMI) significantly differed between the non-obese with non-severe OSA (nO-nS) and obese with non-severe OSA (O-nS) subgroups, between the nO-nS and obese with severe OSA (O-S) subgroups, between the non-obese with severe OSA (nO-S) and O-nS subgroups, and between the nO-S and O-S subgroups at baseline. After AT, BMI z-score significantly increased in the nO-S subgroup. However, post-AT BMI z-score significantly differed between the nO-nS and O-nS subgroups, between the nO-nS and O-S subgroups, between the nO-S and O-nS subgroups, and between the nO-S and O-S subgroups. (**b**) The obstructive apnea-hypopnea index (OAHI) significantly differed between the nO-nS and nO-S subgroups, between the nO-nS and O-S subgroups, between the nO-S and O-nS subgroups, and between the O-nS and O-S subgroups at baseline. After AT, OAHI significantly decreased in the nO-nS, nO-S, and O-S subgroups. Therefore, post-AT AHI only significantly differed between the nO-nS and O-nS subgroups. (**c**) The obstructive apnea index (OAI) significantly differed between the nO-nS and nO-S subgroups and between the nO-S and O-nS subgroups at baseline. After AT, OAI significantly decreased in the nO-nS and O-S subgroups. However, post-AT AHIs were equal across the four subgroups. (**d**) The levels of interleukin (IL)-6 significantly differed between the nO-nS and O-S subgroups, and the O-nS and O-S subgroups at baseline. After AT, the intra-group and inter-group differences in the levels of IL-6 were not statistically significant. (**e**) The levels of IL-9 significantly differed between the nO-nS and O-nS subgroups at baseline. After AT, the intra-group and inter-group differences in the levels of IL-9 were not statistically significant. (**f**) The levels of basic fibroblast growth factor (FGF) significantly differed between the nO-S and O-nS subgroups. After AT, the intra-group and inter-group differences in the levels of basic FGF were not statistically significant. (**g**) The levels of platelet-derived growth factor (PDGF)-BB significantly differed between the nO-S and O-nS subgroups. After AT, PDGF-BB significantly decreased in the nO-nS and nO-S subgroups. However, the inter-group differences in the levels of basic FGF after AT were not statistically significant. (**h**) The levels of regulated on activation, normal T cell expressed and secreted (RANTES) significantly differed between the nO-S and O-nS subgroups. After AT, RANTES significantly decreased in the nO-nS, nO-S, and O-S subgroups. However, the inter-group differences in the levels of RANTES after AT were not statistically significant.

**Figure 2 jcm-09-00579-f002:**
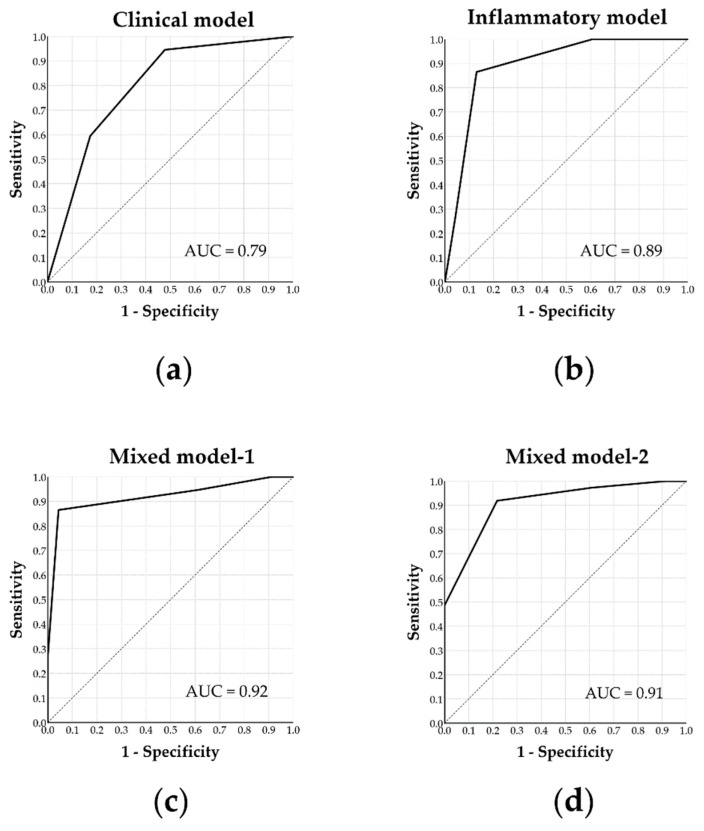
Receiver operator curves using various models to predict surgical cure of adenotonsillectomy in children with obstructive sleep apnea. (**a**) The ‘clinical model’ included age and body mass index z-score to independently predict surgical cure. (**b**) The ‘inflammatory model’ included four independent inflammatory markers (interleukin-1ra, interleukin-17, granulocyte-macrophage colony-stimulating factor, and monocyte chemotactic protein-1) to predict surgical cure. (**c**) The ‘mixed model-1′ included age, granulocyte-macrophage colony-stimulating factor, monocyte chemotactic protein-1, as well as regulated on activation, normal T cell expressed and secreted to best predict surgical cure. (**d**) The ‘Mixed model-2′ included age, body mass index z-score, monocyte chemotactic protein-1, and regulated on activation, normal T cell expressed and secreted to second-best predict surgical cure. **AUC**: area under the curve.

**Table 1 jcm-09-00579-t001:** Patients’ characteristics across subgroups at baseline.

Variables	nO-nS	nO-S	O-nS	O-S	*p*-Value ^1^
Patients	*n* = 24	*n* = 11	*n* = 14	*n* = 11	
Age (years)	6.8 (1.7)	7.6 (2.5)	8.3 (2.5)	7.6 (2.4)	0.26
Males	16 (67)	9 (82)	13 (93)	8 (73)	0.30
BMI (kg/m^2^) z-score	**−0.69 (2.29)** ^2^	**0.03 (1.32)** ^3^	**1.92 (0.51)** ^2,3^	**2.33 (0.54)** ^2,3^	**<0.001**
OSA-18	78.8 (14.6)	82.8 (18.4)	76.6 (15.6)	88.0 (13.8)	0.27
OAHI (events/h)	**3.7 (2.6)** ^2^	**22.3 (14.4)** ^2,3^	**4.9 (2.8)** ^3,4^	**26.1 (16.7)** ^2,4^	**<0.001**
OAI (events/h)	**0.5 (0.6)** ^2^	**6.9 (11.8)** ^2,3^	**0.9 (1.3)** ^3^	5.0 (5.6)	**0.01**
Mean SpO_2_ (%)	**97.7 (0.8) ^2^**	96.3 (2.6)	**97.5 (0.8) ^4^**	**96.6 (0.8) ^2,4^**	**0.02**
Minimal SpO_2_ (%)	**92.0 (2.4) ^2^**	**82.8 (10.7) ^2,3^**	**89.3 (3.1) ^3,4^**	**82.4 (6.6) ^2,4^**	**<0.001**

Data are summarized as mean (standard deviation) or *n* (%) as appropriate. Abbreviations: **AHI**: apnea–hypopnea index; **BMI**: body mass index; **nO-nS**: non-obese with non-severe **OAHI**: obstructive apnea-hypopnea index; O**AI**: obstructive apnea index; OSA; **nO-S**: non-obese with severe OSA; **O-nS**: obese with non-severe OSA; **O-S**: obese with severe OSA; **OSA**: obstructive sleep apnea; **SpO_2_**: oxygen saturation measured by pulse oximetry. ^1^ Data were compared using one-way analysis of variance with post-hoc Tukey’s honestly significant difference tests. ^2^
*p*-value < 0.05 when the variable in the nO-nS subgroup was compared with the nO-S, O-nS, or O-S subgroup. ^3^
*p*-value < 0.05 when the variable in the nO-S subgroup was compared with the O-nS or O-S subgroup. ^4^
*p*-value < 0.05 when the variable in the O-nS subgroup was compared with the O-S subgroup. Significant *p*-values are marked in bold.

**Table 2 jcm-09-00579-t002:** Inflammatory biomarkers across subgroups at baseline.

Variables	nO-nS	nO-S	O-nS	O-S	*p*-Value ^1^
Patients	*n* = 24	*n* = 11	*n* = 14	*n* = 11	
IL-1β (pg/mL)	0.7 (0.6)	0.8 (0.9)	0.5 (0.8)	1.0 (0.9)	0.45
IL-1ra (pg/mL)	131.3 (69.1)	157.0 (83.9)	195.9 (231.0)	223.9 (128.5)	0.25
IL-2 (pg/mL)	3.5 (2.0)	3.8 (2.6)	1.9 (2.1)	3.2 (2.1)	0.13
IL-4 (pg/mL)	3.0 (1.2)	3.1 (0.9)	2.0 (1.4)	2.4 (1.2)	0.07
IL-5 (pg/mL)	11.9 (9.0)	14.7 (22.9)	5.0 (7.8)	20.2 (23.0)	0.11
IL-6 (pg/mL)	**1.8 (1.3)** ^2^	1.7 (1.5)	**1.1 (1.1)** ^4^	**5.4 (8.1)** ^2,4^	**0.02**
IL-7 (pg/mL)	8.6 (5.7)	8.2 (5.4)	6.4 (6.9)	5.9 (3.6)	0.47
IL-8 (pg/mL)	7.7 (4.9)	6.1 (1.9)	5.1 (2.8)	5.7 (1.7)	0.16
IL-9 (pg/mL)	**112.0 (43.4)** ^2^	112.6 (44.8)	**62.8 (65.1)** ^2^	83.0 (47.5)	**0.02**
IL-10 (pg/mL)	1.4 (2.2)	1.1 (2.8)	1.2 (2.0)	0.9 (1.6)	0.94
IL-12 (pg/mL)	2.0 (2.6)	3.4 (8.4)	0.7 (0.9)	2.6 (3.9)	0.47
IL-13 (pg/mL)	1.3 (2.7)	0.8 (1.0)	1.0 (1.4)	1.1 (1.4)	0.93
IL-15 (pg/mL)	44.0 (55.0)	50.3 (51.5)	33.5 (29.1)	54.3 (51.5)	0.73
IL-17 (pg/mL)	17.3 (9.0)	17.9 (8.4)	10.2 (9.7)	18.1 (9.3)	0.08
Eotaxin (pg/mL)	68.1 (43.5)	72.1 (34.6)	61.7 (39.4)	64.3 (34.0)	0.92
Basic-FGF (pg/mL)	**32.0 (8.6)** ^2^	31.5 (11.3)	**20.6 (13.7)** ^2^	29.0 (14.1)	**0.03**
G-CSF (pg/mL)	108.7 (87.5)	102.5 (88.0)	45.0 (77.2)	39.7 (75.2)	0.05
GM-CSF (pg/mL)	1.5 (1.9)	1.2 (2.1)	0.7 (1.0)	2.4 (1.9)	0.15
Interferon-γ (pg/mL)	5.4 (4.8)	6.3 (6.6)	3.9 (5.0)	8.7 (5.5)	0.16
IP-10 (pg/mL)	920.1 (625.6)	733.5 (389.6)	1780.4 (4034.5)	1330.7 (1135.6)	0.54
MCP-1 (pg/mL)	38.5 (16.7)	36.7 (17.2)	39.2 (47.6)	31.3 (10.8)	0.88
MIP-1α (pg/mL)	2.1 (1.3)	1.9 (1.0)	1.5 (0.9)	1.9 (0.7)	0.55
MIP-1β (pg/mL)	135.5 (36.1)	126.7 (28.1)	117.1 (24.2)	128.2 (25.8)	0.36
PDGF-BB (pg/mL)	7490.6 (2432.8)	**8283.6 (2713.4)** ^3^	**5287.8 (3248.3)** ^3^	5821.6 (2592.6)	**0.02**
RANTES (pg/mL)	23757.4 (10285.4)	**26681.1 (9835.8)** ^3^	**15223.4 (7847.7)** ^3^	22973.0 (10319.2)	**0.02**
TNF-α (pg/mL)	45.9 (19.2)	44.9 (19.3)	38.8 (22.8)	47.9 (18.9)	0.67
VEGF (pg/mL)	60.8 (91.5)	33.5 (45.6)	43.2 (28.7)	25.4 (33.6)	0.43

Data are summarized as means (standard deviations). Abbreviations: **FGF**: fibroblast growth factor; **G-CSF**: granulocyte-colony stimulating factor; **GM-CSF**: granulocyte-macrophage colony-stimulating factor; **IL**: interleukin; **IP**: interferon gamma-induced protein; **MCP**: monocyte chemotactic protein; **MIP**: macrophage inflammatory protein; **nO-nS**: non-obese with non-severe OSA; **nO-S**: non-obese with severe OSA; **O-nS**: obese with non-severe OSA; **O-S**: obese with severe OSA; **OSA**: obstructive sleep apnea; **PDGF**: platelet-derived growth factor; **RANTES**: regulated on activation, normal T cell expressed and secreted; **TNF**: tumor necrosis factor; **VEGF**: vascular endothelial growth factor. ^1^ Data were compared using one-way analysis of variance with post-hoc Tukey’s honestly significant difference tests. ^2^
*p*-value < 0.05 when the variable in the nO-nS subgroup was compared with the nO-S, O-nS, or O-S subgroup. ^3^
*p*-value < 0.05 when the variable in the nO-S subgroup was compared with the O-nS or O-S subgroup. ^4^
*p*-value < 0.05 when the variable in the O-nS subgroup was compared with the O-S subgroup. Significant *p*-values are marked in bold.

**Table 3 jcm-09-00579-t003:** Patients’ characteristics across subgroups at least 3 months after AT.

Variables	nO-nS	nO-S	O-nS	O-S	*p*-Value ^1^
Patients	*n* = 24	*n* = 11	*n* = 14	*n* = 11	
BMI (kg/m^2^) z-score	**−0.26 (1.40)** ^2^	**0.51 (1.29)** ^3,^*	**2.06 (0.35)** ^2,3^	**2.20 (0.67)** ^2,3^	**<0.001**
OSA-18	52.2 (13.0) *	55.3 (10.9) *	51.0 (13.2) *	51.1 (11.6) *	0.83
OAHI (events/h)	**1.3 (1.0)** 2,*	1.7 (2.4) *	**3.9 (3.9)** 2	2.4 (3.1) *	**0.03**
OAI (events/h)	0.2 (0.2) *	0.1 (0.1)	0.7 (1.4)	0.4 (0.5) *	0.09
Surgical cure	18 (75%)	9 (82%)	5 (36%)	5 (46%)	**0.03**
Mean SpO_2_ (%)	97.8 (0.8) *	97.6 (1.0)	97.8 (0.6)	97.4 (0.8) *	0.51
Minimal SpO_2_ (%)	92.3 (2.5) *	90.2 (3.8) *	89.6 (2.8)	89.6 (5.4) *	0.10

Data are summarized as mean (standard deviation) or *n* (%) as appropriate. Abbreviations: **AHI**: apnea–hypopnea index; **AI**: apnea index; **BMI**: body mass index; **nO-nS**: non-obese with non-severe OSA; **nO-S**: non-obese with severe OSA; **O-nS**: obese with non-severe OSA; **O-S**: obese with severe OSA; **OSA**: obstructive sleep apnea; **SpO_2_**: oxygen saturation measured by pulse oximetry. ^1^ Data were compared using one-way analysis of variance with post-hoc Tukey’s honestly significant difference tests. ^2^
*p*-value < 0.05 when the variable in the nO-nS subgroup was compared with the nO-S, O-nS, or O-S subgroup. ^3^
*p*-value < 0.05 when the variable in the nO-S subgroup was compared with the O-nS or O-S subgroup. ^4^
*p*-value < 0.05 when the variable in the O-nS subgroup was compared with the O-S subgroup. Significant *p*-values are marked in bold. * *p*-value < 0.05 when variables were compared between baseline and post-AT data.

**Table 4 jcm-09-00579-t004:** Inflammatory biomarkers across subgroups at least 3 months after AT.

Variables	nO-nS	nO-S	O-nS	O-S	*p*-Value ^1^
Patients	*n* = 24	*n* = 11	*n* = 14	*n* = 11	
IL-1β (pg/mL)	0.3 (0.2) *	0.3 (0.2)	0.3 (0.2)	0.3 (0.3) *	0.83
IL-1ra (pg/mL)	93.6 (48.5) *	94.7 (66.7) *	146.3 (99.8)	132.9 (76.4) *	0.11
IL-2 (pg/mL)	2.1 (1.4) *	1.9 (1.3)	1.5 (1.2)	2.1 (1.6)	0.51
IL-4 (pg/mL)	2.0 (1.0) *	1.8 (1.0) *	2.0 (1.0)	1.5 (0.9) *	0.49
IL-5 (pg/mL)	4.1 (4.1) *	5.1 (9.7)	4.1 (4.3)	13.5 (23.9)	0.13
IL-6 (pg/mL)	1.7 (3.1)	0.8 (0.9)	1.1 (0.8)	2.9 (4.9)	0.35
IL-7 (pg/mL)	8.0 (5.7)	9.0 (7.4)	7.9 (6.5)	3.4 (2.3)	0.11
IL-8 (pg/mL)	4.3 (2.3) *	4.4 (2.2)	3.6 (1.7)	4.1 (2.7)	0.77
IL-9 (pg/mL)	98.5 (36.6)	87.5 (56.3)	94.6 (56.3)	68.6 (44.0)	0.36
IL-10 (pg/mL)	0.4 (0.2) *	0.4 (0.2)	0.4 (0.2)	0.3 (0.2)	0.29
IL-12 (pg/mL)	0.9 (1.1)	2.2 (4.8)	0.4 (0.4)	2.5 (4.7)	0.22
IL-13 (pg/mL)	0.7 (0.7) *	0.7 (0.9)	0.8 (1.0)	0.8 (1.0)	0.97
IL-15 (pg/mL)	23.2 (48.8) *	21.7 (14.8)	16.0 (14.1) *	38.8 (70.1)	0.63
IL-17 (pg/mL)	11.7 (6.5) *	9.0 (5.6) *	10.9 (7.0)	13.3 (9.0)	0.54
Eotaxin (pg/mL)	46.2 (20.9) *	44.7 (16.1) *	50.2 (22.2)	48.6 (28.2) *	0.92
Basic-FGF (pg/mL)	29.0 (14.1)	29.5 (17.2)	24.6 (15.6)	23.0 (19.6)	0.65
G-CSF (pg/mL)	84.6 (84.3)	69.0 (77.5) *	82.2 (90.1)	18.0 (36.8)	0.12
GM-CSF (pg/mL)	0.6 (0.9) *	0.5 (0.6)	0.4 (0.4)	1.9 (4.0)	0.18
Interferon-γ (pg/mL)	4.0 (3.3)	3.2 (2.9)	3.5 (2.8)	7.6 (11.7)	0.20
IP-10 (pg/mL)	630.8 (485.9) *	496.1 (288.9) *	564.6 (301.9)	740.4 (392.6)	0.52
MCP-1 (pg/mL)	17.9 (9.5) *	15.2 (8.1) *	21.1 (12.0)	16.8 (6.6) *	0.46
MIP-1α (pg/mL)	1.5 (1.7) *	1.8 (1.3)	1.1 (0.6)	1.5 (0.7)	0.61
MIP-1β (pg/mL)	120.9 (27.2) *	113.3 (18.6)	113.2 (19.0)	124.5 (20.5)	0.51
PDGF-BB (pg/mL)	4816.0 (2312.9) *	6001.3 (2521.4) *	5150.0 (2025.7)	4606.8 (2100.4)	0.45
RANTES (pg/mL)	16651.8 (6150.8) *	14805.6 (5618.9) *	17522.6 (6939.8)	17204.6 (5330.5) *	0.71
TNF-α (pg/mL)	52.8 (24.4)	50.1 (24.4)	42.6 (24.1)	42.9 (19.4)	0.51
VEGF (pg/mL)	38.4 (60.7)	41.1 (34.4)	27.5 (22.2)	45.2 (51.3)	0.81

Data are summarized as means (standard deviations). Abbreviations: **FGF**: fibroblast growth factor; **G-CSF**: granulocyte-colony stimulating factor; **GM-CSF**: granulocyte-macrophage colony-stimulating factor; **IL**: interleukin; **IP**: interferon gamma-induced protein; **MCP**: monocyte chemotactic protein; **MIP**: macrophage inflammatory protein; **nO-nS**: non-obese with non-severe OSA; **nO-S**: non-obese with severe OSA; **O-nS**: obese with non-severe OSA; **O-S**: obese with severe OSA; **OSA**: obstructive sleep apnea; **PDGF**: platelet-derived growth factor; **RANTES**: regulated on activation, normal T cell expressed and secreted; **SD**: standard deviation; **TNF**: tumor necrosis factor; **VEGF**: vascular endothelial growth factor. ^1^ Data were compared using one-way analysis of variance with post-hoc Tukey’s honestly significant difference tests. ^2^
*p*-value < 0.05 when the variable in the nO-nS subgroup was compared with the nO-S, O-nS, or O-S subgroup. ^3^
*p*-value < 0.05 when the variable in the nO-S subgroup was compared with the O-nS or O-S subgroup. ^4^
*p*-value < 0.05 when the variable in the O-nS subgroup was compared with the O-S subgroup. Significant *p*-values are marked in bold.

**Table 5 jcm-09-00579-t005:** Single physiological variables or inflammatory biomarkers as predictors of AT surgical cure in the pediatric OSA patients.

Predictors	Cut-off Value	AUC	95% CI	*p*-Value
Clinical variables				
Age (years)	<7.0	0.70	0.57–0.84	**0.01**
Boys	Boy	0.42	0.27–0.56	0.28
BMI (kg/m^2^) z-score	<1.44	0.74	0.61–0.87	**0.002**
OSA-18	>78.5	0.53	0.38–0.68	0.69
OAHI (events/h)	<4.0	0.63	0.49–0.77	0.09
OAI (events/h)	>0.6	0.59	0.44–0.74	0.25
Mean SpO_2_ (%)	>97.6	0.61	0.46–0.75	0.18
Minimum SpO_2_ (%)	>86.5	0.58	0.43–0.73	0.29
Inflammatory biomarkers				
IL-1β (pg/mL)	>0.3	0.56	0.41–0.71	0.41
IL-1ra (pg/mL)	<149.2	0.67	0.53–0.81	**0.03**
IL-2 (pg/mL)	>3.4	0.62	0.48–0.77	0.11
IL-4 (pg/mL)	>1.8	0.62	0.46–0.77	0.14
IL-5 (pg/mL)	>1.0	0.62	0.47–0.77	0.12
IL-6 (pg/mL)	<2.0	0.55	0.40–0.70	0.49
IL-7 (pg/mL)	>4.9	0.59	0.44–0.74	0.24
IL-8 (pg/mL)	>3.0	0.62	0.46–0.77	0.13
IL-9 (pg/mL)	>82.8	0.67	0.52–0.82	**0.03**
IL-10 (pg/mL)	<0.5	0.69	0.55–0.83	**0.01**
IL-12 (pg/mL)	>0.2	0.63	0.48–0.78	0.10
IL-13 (pg/mL)	<1.4	0.58	0.43–0.73	0.31
IL-15 (pg/mL)	<11.8	0.71	0.58–0.84	**0.01**
IL-17 (pg/mL)	>4.7	0.66	0.51–0.81	**0.04**
Eotaxin (pg/mL)	>36.8	0.61	0.45–0.76	0.17
Basic-FGF (pg/mL)	>15.9	0.65	0.50–0.80	0.06
G-CSF (pg/mL)	>69.5	0.64	0.50–0.78	0.07
GM-CSF (pg/mL)	<0.2	0.68	0.54–0.81	**0.02**
Interferon-γ (pg/mL)	>1.9	0.51	0.36–0.66	0.91
IP-10 (pg/mL)	<818.6	0.56	0.41–0.71	0.45
MCP-1 (pg/mL)	<51.2	0.67	0.52–0.82	**0.03**
MIP-1α (pg/mL)	<1.2	0.51	0.36–0.66	0.87
MIP-1β (pg/mL)	<121.6	0.53	0.38–0.68	0.74
PDGF-BB (pg/mL)	<10201.7	0.62	0.46–0.77	0.13
RANTES (pg/mL)	>15435.5	0.71	0.56–0.85	**0.01**
TNF-α (pg/mL)	>27.9	0.65	0.50–0.80	0.06
VEGF (pg/mL)	<42.1	0.70	0.57–0.84	**0.01**

Abbreviations: **AUC**: area under the curve; **BMI**: body mass index; **CI**: confidence interval; **FGF**: fibroblast growth factor; **G-CSF**: granulocyte-colony stimulating factor; **GM-CSF**: granulocyte-macrophage colony-stimulating factor; **IL**: interleukin; **IP**: interferon gamma-induced protein; **MCP**: monocyte chemotactic protein; **MIP**: macrophage inflammatory protein; **OAHI**: apnea–hypopnea index; **OAI**: apnea index; **OSA**: obstructive sleep apnea; **PDGF**: platelet-derived growth factor; **RANTES**: regulated on activation, normal T cell expressed and secreted; **SpO_2_**: oxygen saturation measured by pulse oximetry; **TNF**: tumor necrosis factor; **VEGF**: vascular endothelial growth factor.

**Table 6 jcm-09-00579-t006:** Prediction models of AT surgical cure in the pediatric OSA patients.

	Logistic Regression			Receiver Operator Characteristic Curve		
Predictors	Odds Ratio	95% CI	*p*-Value	Cut-off Value	Sensitivity	Specificity
**Univariate models**						
Age	5.8	1.9–18.2	**0.002**	<7.0	76%	65%
BMI z-score	8.3	2.5–27.1	**<0.001**	<1.44	78%	70%
OSA-18	1.3	0.5–3.6	0.64	>78.5	54%	52%
OAHI	3.6	1.0–12.7	**0.045**	<4.0	43%	83%
OAI	2.0	0.7–5.9	0.19	>0.6	57%	61%
Mean SpO_2_	2.4	0.8–7.2	0.12	>97.6	51%	70%
Minimal SpO_2_	2.1	0.7–6.2	0.19	>86.5	73%	43%
IL-1ra	4.2	1.4–12.7	**0.01**	<149.2	73%	61%
IL-9	5.9	1.7–20.4	**0.01**	>82.8	87%	48%
IL-10	5.1	1.6–15.6	**0.01**	<0.5	73%	65%
IL-15	11.1	2.3–54.2	**0.003**	<11.8	51%	91%
IL-17	19.2	2.2–167.2	**0.01**	>4.7	97%	35%
GM-CSF	6.3	1.6–25.0	**0.01**	<0.2	49%	87%
MCP-1	11.3	2.2–58.7	**0.004**	<51.2	95%	39%
RANTES	9.0	2.4–33.7	**0.001**	>15435.5	89%	52%
VEGF	5.8	1.9–18.2	**0.002**	<42.1	76%	65%
**Clinical model**				≥1	95%	52%
Age < 7.0 years	4.4	1.2–15.2	**0.03**			
BMI z-score < 1.44	6.6	1.9–22.8	**0.003**			
**Inflammatory model**				≥3	87%	87%
IL-1ra < 149.2 pg/mL	8.3	1.3–53.9	**0.03**			
IL-17 > 4.7 pg/mL	43.3	3.7–504.3	**0.003**			
GM-CSF < 0.2 pg/mL	7.9	1.1–58.4	**0.04**			
MCP-1 < 51.2 pg/mL	11.0	1.4–83.6	**0.02**			
**Mixed model-1**				≥3	87%	96%
Age < 7.0 years	11.6	2.3–90.8	0.004			
GM-CSF < 0.2 pg/mL	8.1	1.2–54.9	0.03			
MCP-1 < 51.2 pg/mL	23.8	2.6–229.8	0.01			
RANTES > 15435.5 pg/mL	14.6	2.3–90.8	0.004			
**Mixed model-2**				≥3	92%	78%
Age < 7.0 years	2.1	1.3–56.0	0.02			
BMI z-score < 1.44	2.1	1.4–50.8	0.02			
MCP-1 < 51.2 pg/mL	3.6	2.9–432.5	0.01			
RANTES > 15435.5 pg/mL	3.1	2.7–191.5	0.004			

Abbreviations: **AUC**: area under the curve; **BMI**: body mass index; **CI**: confidence interval; **GM-CSF**: granulocyte-macrophage colony-stimulating factor; **IL**: interleukin; **MCP**: monocyte chemotactic protein; **OAHI**: obstructive apnea–hypopnea index; **OAI**: obstructive apnea index; **OSA**: obstructive sleep apnea; **RANTES**: regulated on activation, normal T cell expressed and secreted; **SpO_2_**: oxygen saturation measured by pulse oximetry; **VEGF**: vascular endothelial growth factor.

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
