# Peer review of "Relationships Among and Predictive Values of Obesity, Inflammation Markers, and Disease Severity in Pediatric Patients with Obstructive Sleep Apnea Before and After Adenotonsillectomy"

_jcm, 2020, doi:10.3390/jcm9020579_

Round 1

Reviewer 1 Report

The authors have addressed my comments. 

Reviewer 2 Report

The manuscript JCM 720883

Relationship among and Predictive Values of obesity,inflammation markers and disease severity in pediatric patients wits OBS before and after tonsillectomy concerns an important and interesting issue evaluating the treatment of OBS. The study is appropriately  designed and the results are well presented. I would like to recommend this manuscript for print in JCM.

This manuscript is a resubmission of an earlier submission. The following is a list of the peer review reports and author responses from that submission.

Round 1

Reviewer 1 Report

The paper by Chuang and colleagues is a well written paper evaluating obstructive sleep apnea (OSA), the impact of obesity and the impact of adenotonsillectomy on specific inflammatory biomarkers. The authors nicely develop a model for predicting surgical response encompassing in addition to known risk factors for residual OSA but the impact of inflammatory markers. 

Prior to publication however there are several concerns specifically to the paper's methodology. 

1) It is not clear why the authors chose their own criteria for severe and non severe OSA. Classically an obstructive AHI > 10 events/hr is described as severe OSA. The authors should revise their cutoffs to reflect this

2) The authors should use an obstructive AHI rather than an AHI which also includes a central events

3) The authors defined surgical success using their own criteria for success that has been previously published. Notwithstanding, they should also evaluate patients who had resolution of OSA (OAHI<1/hr) as done in previous published studies such as the CHAT study (Marcus CL, et al; Childhood Adenotonsillectomy Trial (CHAT). A randomized trial of adenotonsillectomy for childhood sleep apnea. N Engl J Med. 2013 Jun 20;368(25):2366-76.)

4) The authors characterization is based on arbitrary groups that based on their own criteria. In addition to using more clinically accepting criteria for OSA (OAHI>10/hr for severe), they should also evaluate inflammatory markers of significance against continuous variables such as BMI or OAHI and consider including graphs of these correlations

5) The authors should state in their limitations that there was a predominance of boys in their study population which does not reflect true OSA demographics in children

6) The authors should also report race and ethnicity and acknowledge this as a limitation if they have a limited racial demographic. 

Minor

Spelling mistake on Pg 14 - postulate is spelled postulate. 

Reviewer 2 Report

The authors have comprehensively examined the relationship of obesity, OSA severity and markers of inflammation prospectively before and after adenotonsillectomy in 60 children.

Have few comments for consideration

-Markers of inflammation could be elevated in those patients with chronic diseases, however  such patients were not excluded. How could the confounding co-morbid conditions be taken into consideration?

- what is the average time duration of the sleep study and collection of the markers of inflammation at baseline and during follow- up

- line 406, correct spelling of postulate

- figure 2, not clear especially the labeling